# Pre-labor rupture of membranes and associated factors among pregnant women admitted to the maternity ward, Northwest Ethiopia

**Animut Takele Telayneh**[1]*, **Daniel Bekele Ketema**[1,2], **Belayneh Mengist**[1], **Lieltework Yismaw**[1], **Yibelu Bazezew**[3], **Molla Yigzaw Birhanu**[1], **Samuel Derbie Habtegiorgis**[1]

1 Department of Public Health, Debre Markos University, Debre Markos, Ethiopia, 2 The George Institute for Global Health, University of New South Wales (UNSW), Sydney, Australia, 3 Department of Midwifery, Debre Markos University, Debre Markos, Ethiopia

* animuttakele@gmail.com

**Data Availability Statement:** The datasets used and/or analyzed during the current study are available as supplementary information.

## Abstract

Pre-labor rupture of membranes (PROM) is the rupture of fetal membranes before the onset of labor. PROM is found in 3–15% of all pregnancies and 30–40% of preterm labor worldwide. The most serious complications are neonatal and prenatal mortality, which is higher in Africa, including Ethiopia. Despite a paucity of evidence on the magnitude and factors affecting PROM after 28 weeks of gestation but before the onset of labor (including both term and preterm PROM). Hence, the purpose of this study was to determine the magnitude and identify associated factors of the pre-labor rupture of membranes. An institutional-based cross-sectional study was conducted among 315 pregnant women from April 10, 2019 to June 30, 2019 at Debre Markos Referral Hospital. The samples were chosen using a systematic random sampling method among admitted pregnant women. The data were entered using Epi-Data entry version 4.2 and cleaned and analyzed using Stata/SE version 14.0. In binary logistic regressions, variables with a p-value <0.20 are selected for multivariable analysis. A multivariable logistic regression model with a 95% confidence interval and a p-value <0.05 was used to identify associated factors. In this study, the magnitude of PROM was 19%. Maternal monthly income ≤1000 birr [AOR: 3.33 (95%CI: 1.33, 8.33)], gestational age <37weeks [AOR: 3.28 (95%CI: 1.53, 7.02)], multiple pregnancy [AOR: 4.14 (95%CI: 1.78, 9.62)], polyhydramnios [AOR: 5.06 (95%CI: 2.28, 11.23)] and history of abnormal vaginal discharge [AOR: 6.65 (95%CI: 2.62, 16.72)] were found significant associated factors. In conclusion, the magnitude of the pre-labor rapture of the membranes was higher than in previous studies. Hence, health professionals should strengthen counseling, early diagnosis, and treatment of infections, as well as focus on women with pregnancy-related risks, to reduce pre-labor rupture of membranes and improve fetal and perinatal health.

**Funding:** The author(s) received no specific funding for this work.

**Competing interests:** The authors have declared that no competing interests exist.

## Introduction

Fetal membranes that surround the amniotic cavity retain amniotic fluid, secrete substances, and protect the fetus from infection. Pre-labor rupture of membrane (PROM) is the rupture of fetal membranes after 28 weeks of gestation but before the onset of labor [1–4]. PROM that occurs before 37 weeks of gestation is referred to as preterm premature rupture of membranes (PPROM), and PROM that occurs after 37 weeks is called term PROM [1–6]. PROM affects 3–15% of all pregnancies globally, 30–40% of preterm labor, and 8–10% of term labor [1,6–12]. PROM affects 12.1% of mothers in the United States of America (USA) [13] and 7.7% of primiparous mothers and 10.4% of multiparous mothers in India [14]. Previous studies reported that PPROM is 3.1% in Brazil [15], 2.7% in China [16], and 4.3% in Canada [17]. According to the African literature, 1.3% [18] to 3.3% [19] in Nigeria, 2.4% to 4.7% in Egypt [6], and 13.8% in Uganda pregnant mothers reported PROM. Similarly, 14.6% in Jimma [10], 23.5% in Harar [20], and 13.4% in Nekemte town [21] pregnant mothers had PROM whereas 4.1% in Mizan-Aman Hospitals [22], 13.7% in Debre Tabor General Hospital [23], and 1.4% in Addis Ababa [24] had either preterm or term PROM in Ethiopia. According to contemporary PROM diagnosis and management guidelines, the risk of recurrence of preterm PROM is 16% to 32% [3,25].

PPROM affects 2–20% of all pregnancies and accounts for 18–20% of neonatal deaths [3,18]. PROM neonatal mortality ranges from 2–4% to 7–20% in singleton and twin pregnancies, respectively [3,25]. According to the evidence, PROM caused 10% of preterm and 11% of term deliveries in the USA, as well as 40–75% of neonatal deaths [10,12]. Prematurity is the leading cause of perinatal morbidity and mortality in PPROM [4,26]. PROM is the leading cause of premature birth, accounting for, 18–20% in the USA, 30–40% in Oman, and 30% in Egypt [3,6] premature births. The survey conducted in Iran, prolonged PROM increases neonatal death by tenfold [8]. In a study conducted in Ethiopia, prenatal mortality accounts for 107 per 1000 live births [24].

In the majority of cases, the mechanism of PROM cause is unknown, but it may be related to a structural defect in the membranes caused by collagen deficiency or malformation, which causes the membranes to weaken and be destroyed by the enzymatic process in inflammatory or infectious processes. It is also linked to mechanical factors, such as twin pregnancies caused by uterine volume distention [6,15]. Based on previous evidence, low family income, maternal age, employment, education level, multiple pregnancies, gestational ages, hypertension, diabetes mellitus, history of abortion, abnormal vaginal discharge, ANC follow-up, urinary tract infection, and history of Chorioamnionitis were risk factors for PROM [2,6,15,16,21,22,27–30]. In Ethiopia, sufficient evidence has been published on PROM. However, the majority of previous findings report PROM before or after 37 weeks of gestational age, with only a few studies including pregnant women with completed 28 weeks of gestation but before labor onset. Published data are scarce on the magnitude of PROM from after 28 weeks of gestation to before the onset of labor (both preterm and term PROM) and the factors affecting. This result may help to the reduction of fetal and perinatal complications, reduction of maternal mortality associated with pregnancy risks, and strengthen service integrations and the design of facility-specific interventions through health promotion and counseling for pregnant women, all of which will result in improved maternal healthcare services in healthcare settings. Therefore, the purpose of this study was to determine the magnitude of PROM and identify associated factors in Debre Markos Referral Hospital.

## Materials and methods

### Ethics statement

All the procedures in the present study were approved by Debre Markos University, College of Health Science ethical review committee with reference number C/B/E/Co/68/01/2019 on

April 10, 2019. Verbal informed consent was obtained from each study participant before initiating the study. Confidentiality of the information was maintained. All methods were performed in accordance with the relevant guidelines and regulations.

## Study design and eligibility criteria

An institutional-based cross-sectional study was conducted from April 10, 2019 to June 30, 2019 at Debre Markos Referral Hospital, East Gojjam Zone, Amhara National Regional State. It is 265 Kilometers (km) from Bahir Dar and 300 km from the capital city of Ethiopia, Addis Ababa. Under the Debre Markos Referral Hospital, 102 Health Centers, 8 District Hospitals, and 1 General Hospital served as a referral for over 3.5 million people in the catchment area. According to the hospital report, 61,620 mothers were served in the maternity ward from January to March 2019, with 2,312 pregnant women receiving delivery services. Ultrasounds, laboratory tests, clinical examinations, and physical examinations were performed in the hospital to diagnose gestational age, polyhydramnios, fetal positioning, infections, and pre-existing health conditions of the mothers [31]. All pregnant women admitted to the maternity ward for delivery or contemporary management services were included in the source population. Pregnant women who became critically ill during the data collection period were excluded from this study.

## Sample size determination

We tried to estimate sample size using the single proportion formula and double population formula from previous studies in Ethiopia; the larger sample size was considered the final sample size. A double population formula study conducted in Mekele City public hospitals in 2018 [27] was used to calculate the sample size. The Epi Info$^{TM}$ version 7 software was used to calculate sample size using the following assumptions: confidence interval (CI) = 95%, power = 80%, the proportion of unexposed (history of cesarean section) = 3.4%, odds ratio (OR) = 4.24, level of precision = 5%, and non-response rate = 10% yielding a total of 326 study participants.

## Sampling procedure

Among pregnant mothers admitted to the maternity ward, the systemic random sampling technique was used to select study participants. According to the recent three months of pregnant women, 2,312 received delivery services in the hospital from January to March 2019. The K-interval was calculated using this data, and the first study participant was chosen at random, after which every seventh pregnant woman was interviewed in this study until our sample size was reached.

## Data collection procedure and quality assurance

To ensure consistency, the data collection tool was prepared in English, translated into the local Amharic language, and back to English for analysis. Data were collected using face-to-face interviews administered questionnaires (socio-demographic variables) and maternal chart review (clinical variables i.e. PROM status, gestational age, fetal positioning, multiple pregnancies, and other important diagnostic tests). PROM can be confirmed by a sterile speculum examination of clear vaginal fluid, an alkaline PH test of vaginal fluid, arborization (ferning) of dried vaginal fluid, and an adjunct ultrasound assessment of oligohydramnios after rupture [1–3,10,12,26]. Three-degree midwifery data collectors and one master midwifery supervisor participated. Data quality was maintained by giving one day of training to data collectors and supervisors, pre-testing 5% of the samples at Lumamie District Hospital before actual data

collection began, and close supervision was maintained throughout the data collection period. All completed questionnaires were checked for completeness, clarity, and consistency. Any missing or incomplete data were immediately corrected, and all collected data were reviewed and checked for completeness before data entry.

## Data processing and analysis

The data were coded and entered into EpiData Entry Client version 4.2 and exported to Stata/ SE version 14.0 for data cleaning and analysis. Descriptive statistics are presented using mean, standard deviation (SD), and frequency tables. Variables at binary logistic regression with a p-value <0.20 were selected for the multivariable logistic regression model. To assess model fitness, The Hosmer-Lemeshow goodness of fit test was used. Finally, variables with a p-value <0.05 at 95% CI and corresponding AOR were considered statistically significant factors for PROM.

## Results

### Socio-demographic and economic characteristics

A total of 315 study participants were enrolled in the study, with a 97% response rate. The mean (SD) age of respondents was 31 ± 5.34 years. The majority (95.5%) of study participants were of Amhara ethnicity, 134(42.5%) were educated, and 170(54%) lived in rural areas (**Table 1**).

### Obstetrics and gynecological-related characteristics

In this study, 179(56.8%) of study participants were at term with a gestational age between 37 and 40 weeks, while only 36(11.5%) had five or more multigravidas. One-tenth of pregnant women have a non-cephalic fetal presentation, 87(27.6%) have not had an ANC follow-up, and 61(26.7%) have had an abortion history. The majority of PROM women (93.4%) had no sexual intercourse within seven days before membrane rupture, 53(88.4%) did not perform a pelvic examination before membrane rupture, and 54(90%) were unaware of any causes of membrane rupture (**Table 2**).

### History of medical illness and clinical related characteristics

Only 36(11.4%) of those in the study had a history of abnormal vaginal discharge. Chronic illness was diagnosed in 55(17.5%) pregnant women. The majority of these (72.7%) had hypertension cases, and only 5(5.5%) mothers with chronic illnesses were not treated with medication (**Table 3**).

### Magnitude of Pre-labor rupture of membranes

In this study, 60(19%) pregnant women were diagnosed with a pre-labor rupture of membranes, with the majority 35(11.1%) having rupture of membranes before 37 weeks of gestation (preterm PROM) (Fig 1).

### Associated factors of pre-labor rupture of membranes

In the multivariable regression model, mothers who had low monthly income were 3.33 times more likely to occur PROM compared to those mothers with high monthly income [AOR: 3.33 (95% CI: 1.33, 8.33)]. Mothers with gestational age <37 weeks were 3.28 times more likely to have PROM than mothers with a gestational age of 37 to 40 weeks [AOR: 3.28 (95% CI: 1.53, 7.02)]. The odds of mothers who had twin pregnancies were more than 4-fold more likely to

**Table 1. Socio-demographic and economic characteristics of PROM and associated factors among pregnant women admitted to the maternity ward at Debre Markos Referral Hospital, 2019.**

| Variables | Frequency (N) | Percent (%) |
|---|---|---|
| **Age** | | |
| ≤20 years | 7 | 2.2 |
| 21–35 years | 236 | 74.9 |
| ≥36 years | 72 | 22.9 |
| **Ethnicity** | | |
| Amhara | 301 | 95.5 |
| Other * | 14 | 4.5 |
| **Religion** | | |
| Orthodox | 297 | 94.3 |
| Muslim | 10 | 3.2 |
| Catholic | 8 | 2.5 |
| **Marital status** | | |
| Unmarried | 22 | 7 |
| Married | 293 | 93 |
| **Occupation** | | |
| Unemployed | 208 | 66 |
| Employed | 107 | 34 |
| **Monthly income** | | |
| ≤1000 Ethiopian birr | 49 | 15.6 |
| 1001–2500 Ethiopian birr | 97 | 30.8 |
| ≥2501 Ethiopian birr | 169 | 53.6 |
| **Education status** | | |
| Not educated | 181 | 57.5 |
| Educated | 134 | 42.5 |
| **Partner education status** | | |
| Not educated | 169 | 53.6 |
| Educated | 146 | 47.4 |
| **Residence** | | |
| Urban | 145 | 46 |
| Rural | 170 | 54 |
| **Distance to the health facility** | | |
| <10 Km | 268 | 85 |
| ≥10 Km | 47 | 15 |

Note

*Oromo and Tigray.

have PROM compared to singleton pregnancies [AOR: 4.14 (95% CI: 1.78, 9.62)]. Mothers with polyhydramnios were 5 times more likely to occur PROM compared to non-polyhydramnios [AOR: 5.06 (95% CI: 2.28, 11.23)]. The odds of pregnant mothers who had a history of abnormal vaginal discharge were 6.65 times more likely to have PROM compared to those who had no history of abnormal vaginal discharge [AOR: 6.65 (95% CI: 2.65, 16.72)] (**Table 4**).

## Discussion

The study was designed to determine the magnitude of PROM among pregnant women admitted to Debre Markos Referral Hospital. PROM is a leading cause of unidentified preterm

**Table 2. Obstetrics and gynecologic-related characteristics of PROM among pregnant women admitted to the maternity ward at Debre Markos Referral Hospital, 2019.**

| Variables | Frequency (N) | Percent (%) |
|---|---|---|
| **Gestational age** | | |
| <37 weeks | 103 | 32.7 |
| 37–40 weeks | 179 | 56.8 |
| ≥41 weeks | 33 | 10.5 |
| **Gravida** | | |
| 1 | 71 | 22.5 |
| 2–4 | 208 | 66 |
| ≥5 | 36 | 11.5 |
| **History of amniotic leakage (231)** | | |
| No | 181 | 78.4 |
| Yes | 50 | 21.6 |
| **History of abortion (231)** | | |
| No | 170 | 73.6 |
| Yes | 61 | 26.4 |
| **Frequency of abortion history (61)** | | |
| One time | 43 | 70.5 |
| Two times | 13 | 23.5 |
| Three times and above | 5 | 8.2 |
| **ANC follow-up** | | |
| No | 87 | 27.6 |
| Yes | 228 | 72.4 |
| **Number of ANC visits (228)** | | |
| One time | 16 | 7 |
| Two times | 28 | 12.3 |
| Three times | 75 | 32.9 |
| Four times and above | 109 | 47.8 |
| **Multiple pregnancies status** | | |
| Single | 263 | 83.5 |
| Twin | 52 | 16.5 |
| **Polyhydramnios** | | |
| No | 255 | 81 |
| Yes | 60 | 19 |
| **Fetal position** | | |
| Cephalic | 283 | 89.8 |
| Non-cephalic | 32 | 10.2 |
| **Duration of amniotic leakage (60)** | | |
| 1–24 hours | 42 | 70 |
| >24 hours | 18 | 30 |

birth, and a major cause of fetal, maternal, and neonatal morbidity [11,27,30,32]. PROM diagnosis and management are routine tasks for healthcare workers [30]. The magnitude of PROM in this study was 19% (95% CI: 15.06, 23.79) which is comparable to evidence reported 19.7% in Addis Ababa [33] and 23.5% in Harar town [20]. This similarity might be due to the study settings and the inclusion of study participants. This is a higher finding when compared to studies in Tikur Ambesa teaching hospital 1.4% [24], Jimma 7.2% [34], Mizan Aman Hospital 4.1% [22], Nekemte town public hospitals 13.4% [21], and 13.7% Debre Tabor General

**Table 3. History of medical illness and clinical related characteristics of PROM among pregnant women admitted to the maternity ward at Debre Markos Referral Hospital, 2019.**

| Variables | Frequency (N) | Percent (%) |
|---|---|---|
| **History of abnormal vaginal discharge** | | |
| No | 279 | 88.6 |
| Yes | 36 | 11.4 |
| **History of Urinary tract infection** | | |
| No | 235 | 74.6 |
| Yes | 80 | 25.4 |
| **History of fever** | | |
| No | 269 | 85.4 |
| Yes | 46 | 14.6 |
| **History of cough** | | |
| No | 276 | 87.6 |
| Yes | 39 | 12.4 |
| **Did you take medication for any of your previous illness** | | |
| No | 207 | 65.7 |
| Yes | 108 | 34.3 |
| **Type of medication taken (108)** | | |
| Anti-pain | 19 | 17.6 |
| Antibiotic | 89 | 82.4 |
| **Diagnosed chronic illness** | | |
| No | 260 | 82.5 |
| Yes | 55 | 17.5 |
| **Type of diagnosed chronic illness (55)** | | |
| Hypertension | 40 | 72.7 |
| Diabetes millets | 5 | 9.1 |
| CHF | 4 | 7.3 |
| CKD | 2 | 3.6 |
| Epilepsy | 4 | 7.3 |

**Fig 1. Magnitude of PROM among pregnant women admitted to the maternity ward at Debre Markos Referral Hospital.**

**Table 4. Bi-variable and multivariable analysis of PROM and associated factors among pregnant women admitted to the maternity ward at Debre Markos Referral Hospital, 2019.**

| Variables | PROM | | COR (95% CI) | AOR (95% CI) | P-value |
|---|---|---|---|---|---|
| | Yes N (%) | No N (%) | | | |
| **Monthly income** | | | | | |
| ≤1000 birr | 16(5%) | 33(10.5%) | 2.66(1.28, 5.52) | 3.33(1.33, 8.33) | 0.01 |
| 1001–2500 birr | 18(5.7%) | 79(25%) | 1.25(0.64, 2.42) | 0.81(0.34, 1.92) | 0.639 |
| ≥2501 birr | 26(8.2%) | 143(45.4%) | 1 | 1 | |
| **Educational status** | | | | | |
| Not educated | 40(12.7%) | 141(44.8%) | 1.61(0.89, 2.92) | 1.60(0.71, 3.64) | 0.256 |
| Educated | 20(6.3%) | 114(36.2%) | 1 | 1 | |
| **Partner education status** | | | | | |
| Not educated | 38(12%) | 131(41.6%) | 1.63(0.91, 2.92) | 1.40(0.64, 3.06) | 0.393 |
| Educated | 22(7%) | 124(39.4%) | 1 | 1 | |
| **Distance to HF** | | | | | |
| <10km | 45(14.3%) | 223(70.8%) | 1 | 1 | |
| ≥10km | 15(4.7%) | 32(10.2%) | 2.32(1.16, 4.64) | 1.90(0.78, 4.64) | 0.157 |
| **Gestational age** | | | | | |
| <37 weeks | 35(11.1%) | 68(21.6%) | 4.09(2.20, 7.59) | 3.28(1.53, 7.02) | 0.002 |
| 37–40 weeks | 20(6.3%) | 159(50.5%) | 1 | 1 | |
| ≥41 weeks | 5(1.6%) | 28(8.9%) | 1.42(0.49, 4.09) | 0.96(0.26, 3.53) | 0.961 |
| **ANC follow up** | | | | | |
| No | 24(7.6%) | 62(19.7%) | 2.03(1.12, 3.66) | 1.35(0.61–3.01) | 0.455 |
| Yes | 36(11.4%) | 193(61.3%) | 1 | 1 | |
| **Multiple pregnancies** | | | | | |
| Single | 37(11.7%) | 226(71.8%) | 1 | 1 | |
| Twin | 23(7.3%) | 29(9.2%) | 4.84(2.53, 9.26) | 4.14(1.78, 9.62) | 0.001 |
| **Polyhydramnios** | | | | | |
| No | 34(10.8%) | 221(70.2%) | 1 | 1 | |
| Yes | 26(8.2%) | 34(10.8%) | 4.97(2.65, 9.28) | 5.06(2.28, 11.23) | 0.001* |
| **History of abnormal vaginal discharge** | | | | | |
| No | 39(12.4%) | 240(76.2%) | 1 | 1 | |
| Yes | 21(6.6%) | 15(4.8%) | 8.6(4.09, 18.13) | 6.65(2.65, 16.72) | 0.001* |
| **History of urinary tract infection** | | | | | |
| No | 39(12.4%) | 196(62.3%) | 1 | 1 | |
| Yes | 21(6.6%) | 59(18.7%) | 1.78(0.97, 3.27) | 1.76(0.75, 4.13) | 0.189 |
| **History of fever** | | | | | |
| No | 45(14.3%) | 224(71.1%) | 1 | 1 | |
| Yes | 15(4.7) | 31(9.9%) | 1.78(0.97, 3.27) | 1.12(0.39, 3.22) | 0.824 |

Significant level

*<0.001.

Hospital [23]. This is also higher in Nigeria 3.3% [19], Uganda 13.8% [9], Pakistan 3.27% [7], London 15% [35], Spain 1.2% [36], Pakistan 2% [37], Canada 4.3% [17], Iran 7.7% to 10% [8,38], China 2.7% [16], and India 7.69% primiparity and 10.41% multipara [14]. Previous studies included pregnant mothers before or after 37 weeks of gestation [7,16,17,19,22–24], whereas this study includes pregnant women after 28 weeks of gestation but before the onset of labor or both preterm or term PROM. This disparity could be attributed to the difference in study participant inclusion, sample size, or study setting. This finding, however,

is lower than that of the studies conducted in Italy 25.7% [39], Philadelphia 30.7% [40], and Iran 33.7% [41]. The discrepancy could be explained by differences in study settings, sample size, and recruiting gestational age of study participants using non-gold standard diagnosis techniques.

This study revealed various associated factors for the occurrence of PROM. Low-monthly income mothers were 3.33 times more likely to develop PROM compared to high-income mothers. This finding is consistent with previous findings from Ethiopia, Brazil, and Pakistan [2,7,15]. Mothers with low socio-economic status were worried about financial scarcity and were less likely to seek health care services. Hence, they may have poor hygiene practices, recurrent genitourinary infections, stress, inadequate ANC follow-up services, anemia, and poor nutritional status, all of which reduce collagen formation leading to the weakening of membranes and thus increase the likelihood of membrane rupture [2,4].

Likewise, gestational age <37 weeks were 3.28 times more likely to occur PROM compared to gestational age between 37 and 40 weeks. This evidence is similar to previous findings in Uganda, Nigeria, India, Italy, Bangladesh, and China [9,13,19,30,39,40,42,43]. Gestational week <37 pregnancy had probably fetal malpresentation, fetal malposition, and abnormal fetal lie compared to those at 37–40 weeks because the fetus has enough freedom of movement, which may increase the likelihood of direct contact with the weakest part of the membrane [1,9]. In addition, due to hydronephrosis and urinary stasis, pregnant women in late gestational age are less likely to develop urinary tract infections than those in early gestation [9]. This high risk of infection and inflammation in early pregnancy may result in membrane weakening due to poor collagen assembly, altered tensile strength, and collagen breakdown, which stimulates uterine contraction and may result in PROM.

In this study, mothers who had twin pregnancies were more than 4-fold more likely to have PROM compared to singleton counterparts. This finding is consistent with the evidence on contemporary diagnosis and management of PROM [3,4]. The uterus can hold two or more babies; as the fetal membranes become overstretched, the risk of membrane rupture increases.

Mothers with polyhydramnios were 5-fold more likely to have PROM compared to non-polyhydramnios. Amniotic production of interleukin-8 and prostaglandin E embodies biochemical changes in the fetal membranes that are probably commenced by physical forces to overcome force-induced and biochemically induced membrane rupture. This finding is supported by current evidence on the diagnosis and management of PROM [3,4].

Similarly, a history of abnormal vaginal discharge increases the likelihood of PROM by 6.65 times when compared to counterparts. This finding is similar to studies conducted in Ethiopia, Uganda, Nigeria, Pakistan, and Oman [11,23,27,44–46]. The infection causes membrane inflammation, which leads to tensile strength loss, and genital infection by bacteria or other infectious agents stimulates enzyme production like proteases, phospholipases, and collagenases which cause membrane weakness and rupture [27,47]. Infections such as urinary tract infections, sexually transmitted diseases, bacterial vaginosis, and Chorioamnionitis may cause abnormal vaginal discharge. Pregnant women, on the other hand, are at a higher risk of infection due to physiological changes and invasive uterine procedures may increase the risk of pre-labor rupture of membranes.

ANC follow-up, abortion history, urinary tract infection history, and fetal positioning were not statistically associated with PROM in this study, unlike previous findings in Ethiopia [2,21,23,27,28]. However, 27.4% had not received ANC follow-up, 26.4% had an abortion, 25.4% had a urinary tract infection, and 10% had a non-cephalic fetal position, as reported in this finding.

## Limitations

This study was conducted in a single referral hospital and may be difficult to generalize to all health facilities; referrals of high-risk mothers may increase the magnitude; and the truthfulness of the information is dependent on the respondents' responses, which introduces information bias that must be considered in this study.

## Conclusions

In this study, the magnitude of PROM is higher than in previous studies. Maternal low monthly income, <37 weeks of gestational age, multiple pregnancies, polyhydramnios, and a history of abnormal vaginal discharge were identified as statistically significant associated factors. Priority is given to strengthening counseling, early testing, and infection treatment for mothers before and after conception. Health professionals and other responsible bodies should emphasize to pregnant mothers with low socioeconomic status, earlier gestation, and women with pregnancy-related risks to reduce pre-labor rupture of membranes and improve fetal and perinatal health.

## Supporting information

**S1 Data. Pre-labor rupture of membrane data sets among pregnant women admitted to Debre Markos Referral Hospital, 2019.**
(DTA)

## Acknowledgments

We would like to thank Debre Markos University College of Health Science for unreserved support, giving ethical clearance, Debre Markos Referral Hospital staff for their cooperation, willingness to collect data, data collectors, and all others who had to contribute to accomplishing this research work.

## Author Contributions

**Conceptualization:** Animut Takele Telayneh.

**Data curation:** Belayneh Mengist, Molla Yigzaw Birhanu.

**Formal analysis:** Animut Takele Telayneh, Daniel Bekele Ketema, Samuel Derbie Habtegiorgis.

**Investigation:** Yibelu Bazezew, Molla Yigzaw Birhanu.

**Methodology:** Animut Takele Telayneh, Daniel Bekele Ketema, Lieltework Yismaw, Samuel Derbie Habtegiorgis.

**Software:** Animut Takele Telayneh, Daniel Bekele Ketema, Yibelu Bazezew, Samuel Derbie Habtegiorgis.

**Supervision:** Belayneh Mengist, Lieltework Yismaw, Yibelu Bazezew, Molla Yigzaw Birhanu, Samuel Derbie Habtegiorgis.

**Validation:** Lieltework Yismaw, Yibelu Bazezew, Samuel Derbie Habtegiorgis.

**Writing – original draft:** Animut Takele Telayneh, Daniel Bekele Ketema, Belayneh Mengist, Lieltework Yismaw, Yibelu Bazezew, Molla Yigzaw Birhanu, Samuel Derbie Habtegiorgis.

**Writing – review & editing:** Animut Takele Telayneh, Daniel Bekele Ketema, Belayneh Mengist, Yibelu Bazezew, Molla Yigzaw Birhanu, Samuel Derbie Habtegiorgis.

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
