## [Editor Report · Decision Letter 0]

2 May 2022

PGPH-D-22-00688

Pre-labor rupture of membranes and associated factors among pregnant mothers admitted in the maternity ward Northwest Ethiopia

Dear Dr. Animut Takele Telayneh,

Thank you for submitting your manuscript to PLOS Global Public Health. After careful consideration, we feel that it has merit but does not fully meet PLOS Global Public Health’s publication criteria as it currently stands. Therefore, we invite you to submit a revised version of the manuscript that addresses the points raised during the review process.

Please submit your revised manuscript by . If you will need more time than this to complete your revisions, please reply to this message or contact the journal office at globalpubhealth@plos.org. Please include the following items when submitting your revised manuscript:

We look forward to receiving your revised manuscript.

Kind regards,

Thang Van Vo, PhD

Academic Editor

Journal Requirements:

1. Please provide the full name of the institutional review board (IRB) who approved your study.

2. In the online submission form, you indicated that “The datasets used and/or analyzed during the current study are available from the corresponding author upon reasonable request.”. All PLOS journals now require all data underlying the findings described in their manuscript to be freely available to other researchers, either 1. In a public repository, 2. Within the manuscript itself, or 3. Uploaded as supplementary information.

Additional Editor Comments (if provided):

This manuscript is meeting PGPH Journal's publication criteria in the basis and quite interesting to clarify better the problem of PROM and associated factors in obstetric care. Before your manuscript will be sent for peer review, the formula of sample size calculation, in the Sampling procedure section at page 93, should be showed by the formula (not presenting in text) following the cross sectional study design and not used OR for sample estimation. You could use p estimation from previous studies to calculate sample size for your study.
---

## [Decision Letter · Decision Letter 1]

18 Jul 2022

PGPH-D-22-00688R1

Pre-labor rupture of membranes and associated factors among pregnant mothers admitted in the maternity ward Northwest Ethiopia

Dear Dr. Telayneh,

Thank you for submitting your manuscript to PLOS Global Public Health. After careful consideration, we feel that it has merit but does not fully meet PLOS Global Public Health’s publication criteria as it currently stands. Therefore, we invite you to submit a revised version of the manuscript that addresses the points raised during the review process.

We look forward to receiving your revised manuscript.

Kind regards,

Thang Van Vo, PhD

Academic Editor

Journal Requirements:

Additional Editor Comments (if provided):

Dear authors,

Thank you for submitting your manuscript:

It has been reviewed by two experts in the field and we request that you make

major revisions before it is processed further.

Reviewers' comments:

Reviewer's Responses to Questions

**Comments to the Author**

1. If the authors have adequately addressed your comments raised in a previous round of review and you feel that this manuscript is now acceptable for publication, you may indicate that here to bypass the “Comments to the Author” section, enter your conflict of interest statement in the “Confidential to Editor” section, and submit your "Accept" recommendation.

Reviewer #1: (No Response)

Reviewer #2: (No Response)

2. Does this manuscript meet PLOS Global Public Health’s publication criteria? Is the manuscript technically sound, and do the data support the conclusions? The manuscript must describe methodologically and ethically rigorous research with conclusions that are appropriately drawn based on the data presented.

Reviewer #1: No

Reviewer #2: Partly

3. Has the statistical analysis been performed appropriately and rigorously?

Reviewer #1: No

Reviewer #2: Yes

4. Have the authors made all data underlying the findings in their manuscript fully available (please refer to the Data Availability Statement at the start of the manuscript PDF file)?

Reviewer #1: Yes

Reviewer #2: Yes

5. Is the manuscript presented in an intelligible fashion and written in standard English?

Reviewer #1: No

Reviewer #2: No

6. Review Comments to the Author

Reviewer #1: Abstract

Lines 44-45 The sentence should be rewritten: “Hence, we will reduce fetal and prenatal complications by giving attention to pregnant mothers with identified factors”, since the reduction in fetal and/or maternal complications also depend on a range of factors and intervention and not only by “giving attention …”.

Line 38: “The prevalence of PROM in this finding was 19%” should be “The prevalence of PROM in this study was 19%”

Line 43: “pre-labor rapture of the membrane” should be “pre-labor rupture of the membranes”

Line 47: "Author summary?"

Line 52: define the abbreviation of PPROM and make clear the difference between PROM and PPROM.

Methods

- Line 119: “Finally, a total of 326 study participants ...": pls change to “… need to be recruited”?

- In the previous version of the manuscript, the editor commented “the formula of sample size calculation, ..., should be showed by the formula (not presenting in text) following the cross-sectional study design ... You could use p estimation from previous studies to calculate sample size for your study”. While being explained within the response to the editor and reviewers, that requested formula has not been incorporated into the revised version and needs to be inserted.

- Lines 126-127: Are the cases having rupture of membranes confirmed only by interviews conducted by midwives and not by vaginal examination? Is there any test of confirmation for the PROM being implemented? Please elaborate this most important modality and examination for confirmation of PROM in clinical setting.

Results & Discussion

Table 2: The authors presented data on polyhydramnios condition. How was the polyhydramnios assessed and/or confirmed? By ultrasound examination or by obstetrical examination? Quantitative or semi-quantitative modality?

Fetal position could be an influencing factor, abnormal fetal position near term could be a contributing factor to PROM status. Please provide more discussion.

Reviewer #2: I have attached it as a word and I expect the authors fulfill those comments and come back for further review.

7. PLOS authors have the option to publish the peer review history of their article (what does this mean?). If published, this will include your full peer review and any attached files.

**Do you want your identity to be public for this peer review?** For information about this choice, including consent withdrawal, please see our Privacy Policy.

Reviewer #1: No

Reviewer #2: **Yes: **Abela, Abebe Negesso (BSc,MD,MPH)

---

## [Decision Letter · Decision Letter 2]

19 Dec 2022

PGPH-D-22-00688R2

Pre-labor rupture of membranes and associated factors among pregnant women admitted to the maternity ward, Northwest Ethiopia

Dear Dr. Telayneh,

Thank you for submitting your manuscript to PLOS Global Public Health. After careful consideration, we feel that it has merit but does not fully meet PLOS Global Public Health’s publication criteria as it currently stands. Therefore, we invite you to submit a revised version of the manuscript that addresses the points raised during the review process.

We look forward to receiving your revised manuscript.

Kind regards,

Thang Van Vo, PhD

Academic Editor

Journal Requirements:

2. We have noticed that you have uploaded Supporting Information files, but you have not included a list of legends. Please add a full list of legends for your Supporting Information files after the references list.

Additional Editor Comments (if provided):

Dear author.

Sorry for delay in earlier response to your manuscript revisions upon the two reviewers's comments. Before we are going to make a final decision, can you please check and revise with some comments as follows:

1. Check Table 2 to make sure your sample size of study is 315 for data analysis, also check this consistent sample size in overall manuscript (not 326 in total number)

2. Add evidence of ethical approval document in part of Ethical considerations, specify Number and the date of document signed by IRB

Reviewers' comments:

Reviewer's Responses to Questions

**Comments to the Author**

1. If the authors have adequately addressed your comments raised in a previous round of review and you feel that this manuscript is now acceptable for publication, you may indicate that here to bypass the “Comments to the Author” section, enter your conflict of interest statement in the “Confidential to Editor” section, and submit your "Accept" recommendation.

Reviewer #1: (No Response)

Reviewer #2: (No Response)

2. Does this manuscript meet PLOS Global Public Health’s publication criteria? Is the manuscript technically sound, and do the data support the conclusions? The manuscript must describe methodologically and ethically rigorous research with conclusions that are appropriately drawn based on the data presented.

Reviewer #1: Yes

Reviewer #2: Partly

3. Has the statistical analysis been performed appropriately and rigorously?

Reviewer #1: Yes

Reviewer #2: No

4. Have the authors made all data underlying the findings in their manuscript fully available (please refer to the Data Availability Statement at the start of the manuscript PDF file)?

Reviewer #1: Yes

Reviewer #2: Yes

5. Is the manuscript presented in an intelligible fashion and written in standard English?

Reviewer #1: No

Reviewer #2: No

6. Review Comments to the Author

Reviewer #1: The authors have made substantial revisions within the R2 version.

There are still some points of concern that have been raised or newly emerged that need to be solved by the authors:

- Lines 126-135: please elaborate and made a consistent revision to the sample size calculation and the sample size estimate formula (to be added to the manuscript) as the authors mentioned in an appropriate within the Response to the Reviewers section.

- Lines 142-144: “PROM is confirmed clinically (painless gush of fluid leaking from the vagina and a change in color or a decrease in the size of the uterus”: the change in color of what? of the fluid? a decrease in the size of the uterus is not the clinical sign of PROM, unless it’s a severe polyhydramnios case.

- Lines 216-217: “and India 7.69 of primiparity and 10.41 of multipara”: check syntax and “%” character.

- Lines 278-279: “Strengthen counseling, early testing, and treatment including partners for sexually transmitted 279 infections before conception”: check syntax for this sentence.

Reviewer #2: The language part i recommend the manuscript edited by some one whose english language is native or other known editor.

I hope the authors address those major comments i have mentioned on the attached document.

7. PLOS authors have the option to publish the peer review history of their article (what does this mean?). If published, this will include your full peer review and any attached files.

**Do you want your identity to be public for this peer review?** For information about this choice, including consent withdrawal, please see our Privacy Policy.

Reviewer #1: No

Reviewer #2: No

---

## [Decision Letter · Decision Letter 3]

1 Feb 2023

PGPH-D-22-00688R3

Pre-labor rupture of membranes and associated factors among pregnant women admitted to the maternity ward, Northwest Ethiopia

Dear Dr. Telayneh,

Thank you for submitting your manuscript to PLOS Global Public Health. After careful consideration, we feel that it has merit but does not fully meet PLOS Global Public Health’s publication criteria as it currently stands. Therefore, we invite you to submit a revised version of the manuscript that addresses the points raised during the review process.

We look forward to receiving your revised manuscript.

Kind regards,

Dickson Abanimi Amugsi, PhD

Academic Editor

Journal Requirements:

Additional Editor Comments (if provided):

Thank you, for submitting your work to PGPH for publication. It was reviewed by three independent reviewers. Although reviewer 2 recommended that the manuscript should be accepted for publication, reviewer 3 who came onboard due to the inability of reviewer 1 to submit their comments recommended a minor revision. Please, ensure that you adequately address their comments to avoid delay in the publication of your paper.

The issue of language kept coming up. I suggest a critical proofread of the manuscript to address diction and grammar issues. Where possible, you could get a native English speaker to proofread the manuscript.

Thank you

Reviewers' comments:

Reviewer's Responses to Questions

**Comments to the Author**

1. If the authors have adequately addressed your comments raised in a previous round of review and you feel that this manuscript is now acceptable for publication, you may indicate that here to bypass the “Comments to the Author” section, enter your conflict of interest statement in the “Confidential to Editor” section, and submit your "Accept" recommendation.

Reviewer #2: All comments have been addressed

Reviewer #3: All comments have been addressed

2. Does this manuscript meet PLOS Global Public Health’s publication criteria? Is the manuscript technically sound, and do the data support the conclusions? The manuscript must describe methodologically and ethically rigorous research with conclusions that are appropriately drawn based on the data presented.

Reviewer #2: Yes

Reviewer #3: Yes

3. Has the statistical analysis been performed appropriately and rigorously?

Reviewer #2: Yes

Reviewer #3: Yes

4. Have the authors made all data underlying the findings in their manuscript fully available (please refer to the Data Availability Statement at the start of the manuscript PDF file)?

Reviewer #2: Yes

Reviewer #3: Yes

5. Is the manuscript presented in an intelligible fashion and written in standard English?

Reviewer #2: Yes

Reviewer #3: Yes

6. Review Comments to the Author

Reviewer #2: I can't comment better for the language and you may have invite some one native of English

Reviewer #3: Abstract

Introduction - Unclear sentence “Despite a paucity of evidence on the magnitude and the factors affecting PROM after 28 weeks of gestation to before the onset of labor”

Methods

Sample size determination shall be written separately

The sampling procedure shall be explained a bit more mentioning patient flow

How PROM status is asserted in this study is not mentioned. Whether by physical examination or from chart review.

Results

The magnitude of the problem under study is mentioned in Socio-demographic and economic characteristics section and needs to be written separately.

History of medical illness and clinical related factors and Obstetrics and gynecological-related factors shall be presented as characteristics not factors.

The * in the foot note of table 4 is not important as the numbers are mentioned in the column titled p-value

Discussion

It has been mentioned in the first paragraph as the high magnitude of PROM in this study with other studies in Addis Ababa and Harar are due to the type of setting as “This similarity might be due to the study settings, 207 which were conducted in referral hospitals, which increase the high risk of pregnant women being referred from different health facilities for better diagnosis and management.” But also it is mentioned that this finding is much higher than studies in “ Tikur Ambesa 209 teaching hospital 1.4% [24], Jimma 210 7.2% [34], Mizan Aman Hospital 4.1% [22], Nekemte town public hospitals 13.4% [21]” which are also referral hospitals receiving high risk mothers. This makes your justification to the finding questionable.

The study’s finding that those mother with a gestational age 37-40 weeks are less likely to have PROM than those less than 37 weeks of gestation is explained by the statement “When the body prepares for normal labor and delivery, fetal membranes become weakened and fragile.” Do you think this is logical?

7. PLOS authors have the option to publish the peer review history of their article (what does this mean?). If published, this will include your full peer review and any attached files.

**Do you want your identity to be public for this peer review?** For information about this choice, including consent withdrawal, please see our Privacy Policy.

Reviewer #2: No

Reviewer #3: **Yes: **Tomas Yeheyis

---

## [Decision Letter · Decision Letter 4]

16 Feb 2023

Pre-labor rupture of membranes and associated factors among pregnant women admitted to the maternity ward, Northwest Ethiopia

PGPH-D-22-00688R4

Dear Mr Telayneh,

We are pleased to inform you that your manuscript 'Pre-labor rupture of membranes and associated factors among pregnant women admitted to the maternity ward, Northwest Ethiopia' has been provisionally accepted for publication in PLOS Global Public Health.

Best regards,

Dickson Abanimi Amugsi, PhD

Academic Editor

Reviewer Comments (if any, and for reference):

Reviewer's Responses to Questions

**Comments to the Author**

1. If the authors have adequately addressed your comments raised in a previous round of review and you feel that this manuscript is now acceptable for publication, you may indicate that here to bypass the “Comments to the Author” section, enter your conflict of interest statement in the “Confidential to Editor” section, and submit your "Accept" recommendation.

Reviewer #3: All comments have been addressed

2. Does this manuscript meet PLOS Global Public Health’s publication criteria? Is the manuscript technically sound, and do the data support the conclusions? The manuscript must describe methodologically and ethically rigorous research with conclusions that are appropriately drawn based on the data presented.

Reviewer #3: Yes

3. Has the statistical analysis been performed appropriately and rigorously?

Reviewer #3: Yes

4. Have the authors made all data underlying the findings in their manuscript fully available (please refer to the Data Availability Statement at the start of the manuscript PDF file)?

Reviewer #3: Yes

5. Is the manuscript presented in an intelligible fashion and written in standard English?

Reviewer #3: Yes

6. Review Comments to the Author

Reviewer #3: (No Response)

7. PLOS authors have the option to publish the peer review history of their article (what does this mean?). If published, this will include your full peer review and any attached files.

**Do you want your identity to be public for this peer review?** For information about this choice, including consent withdrawal, please see our Privacy Policy.

Reviewer #3: **Yes: **Tomas Yeheyis
